# KG-MoE: Multimodal Knowledge Graph Grounded Mixture of Experts for Fair Visual Question Answering

## Abstract

Mixture-of-Experts architectures scale model capacity efficiently but remain limited by correlation-driven routing, lack of explicit knowledge grounding, and subgroup disparities in high-stakes domains. We propose KG-MoE, a knowledge-based and fairness-aware MoE framework that integrates structured knowledge graphs into expert specialization and employs adversarial debiasing to reduce subgroup risk. A dynamic gating network routes inputs across modality-specific experts while retrieved subgraphs constrain reasoning and guide explanation generation. We derive theoretical bounds showing that knowledge grounding reduces excess risk under distribution shift and that fairness regularization improves worst-group generalization. Empirically, KG-MoE achieves state-of-the-art performance across multimodal benchmarks, including dermoscopic, clinical, and histopathology tasks in dermatology, while reducing demographic parity gaps by more than 50% relative to foundation model baselines. Ablation studies confirm the dual benefit of knowledge integration and fairness constraints for both robustness and equity, and qualitative analysis demonstrates knowledge-based explanations aligned with domain reasoning. Our results position KG-MoE as a general paradigm for trustworthy, interpretable, and fair multimodal learning systems.

## 1 Introduction

Mixture-of-Experts (MoE) architectures scale model capacity by activating only a subset of experts per input, yielding strong accuracy–efficiency trade-offs across modalities (Fedus et al., 2022; Shazeer et al., 2017). Yet contemporary MoE systems remain largely *correlation-driven*: gating decisions rely on data embeddings without structured priors (Mu & Lin, 2025), offering limited guarantees on interpretability, robustness under shift, or subgroup fairness (Sharma et al., 2022). In safety-critical applications, rare conditions, long-tail concepts, and demographic differences can cause brittle gating and uneven errors across subgroups, posing challenges for MoE systems (Zhang et al., 2025). Multimodal settings add complexity by combining multiple data types, which can increase disparities in subgroup performance if fairness measures are not applied (Adewumi et al., 2024; Shang et al., 2024).

Foundation models (FMs) promise broad generalization via large-scale pretraining and multimodal alignment (Bommasani et al., 2021; Wiggins & Tejani, 2022). Nevertheless, FMs are fundamentally data-driven and susceptible to hallucinations when explicit, structured knowledge is absent (Ji et al., 2023); they also lack built-in mechanisms to control subgroup disparities (Xu et al., 2024). These observations motivate architectures that *couple* learned perception with *knowledge-grounded* reasoning *and* fairness-aware training, ideally with theoretical insight into when such coupling reduces risk under distribution shift and improves worst-group performance (Sagawa et al., 2019).

We advance this paradigm with **KG-MoE**, a knowledge-grounded and fairness-aware Mixture-of-Experts framework. A *probabilistic gating function* parameterized over joint feature embeddings and retrieved subgraph representations learns *structured routing distributions*, aligning expert specialization with knowledge-informed contexts. Concretely, retrieved subgraphs from a task-relevant knowledge graph (KG) condition both the gating policy and expert inference, while fairness is promoted via a combination of worst-group risk and adversarial invariance objectives. We also provide theoretical results showing that (i) knowledge conditioning reduces excess risk under covariate shift and (ii) fairness regularization improves worst-group generalization (Sec. 3).

We instantiate KG-MoE in a high-stakes medical setting: *dermatology*, which presents over 3,000 conditions, inherently multimodal diagnostics (clinical photography, dermoscopy, histopathology), and persistent disparities across skin tones (Hay et al., 2014; Giansanti, 2023). Our evaluation spans more than 200k multimodal images and a proprietary curated dermatology VQA corpus of 1.5M validated pairs. KG-MoE achieves state-of-the-art performance while reducing the Fitzpatrick parity gap by over 50% relative to strong FM baselines. Ablations attribute gains to both knowledge grounding and fairness objectives; qualitative analyses show knowledge-aligned explanations that mirror clinical reasoning.

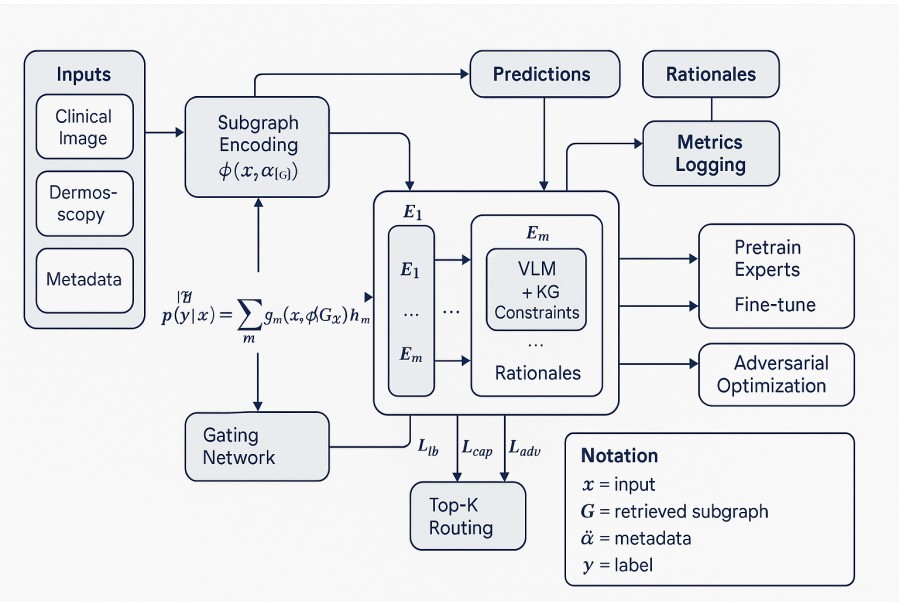

Figure 1: KG-MoE Architecture: The framework integrates knowledge graph retrieval with expert routing. Input images and metadata are processed by modality-specific experts, while retrieved KG subgraphs condition the gating network to select the most relevant experts. The system outputs both predictions and knowledge-grounded explanations.

**Research question.** *Can structured knowledge act as a compass for routing experts fairly in multimodal Mixture-of-Experts models?*

**Contributions.** We propose **KG-MoE**, a Mixture-of-Experts framework that integrates retrieved KG subgraphs into both routing and inference via a probabilistic knowledge-conditioned gating distribution, aligning expert specialization with structured context. To promote subgroup robustness, we introduce a fairness-aware training objective that combines worst-group risk minimization with adversarial invariance, improving equity without sacrificing overall accuracy. Finally, we provide theoretical results showing that knowledge conditioning reduces excess risk under distribution shift while fairness regularization enhances worst-group generalization, and we corroborate these findings with large-scale multimodal experiments in dermatology as a representative high-stakes domain.

## 2 RELATED WORK

The Mixture-of-Experts (MoE) paradigm enables efficient scaling of neural networks through conditional computation. Seminal work on sparsely-gated MoEs demonstrated massive capacity expansion without proportional compute increases (Shazeer et al., 2017). Subsequent advances pushed the scale of these models with GShard enabling massive parallelism (Lepikhin et al., 2020), and the Switch Transformer (Fedus et al., 2022) and GLaM (Du et al., 2022) achieving trillion-parameter scale with high efficiency. Routing mechanisms have also evolved, with methods like balanced assignment (Lewis et al., 2021), hash-based routing (Roller et al., 2021), and Expert Choice routing (Zhou et al., 2022) improving expert utilization and training stability.

To improve model reasoning and factual consistency, researchers have integrated structured knowledge. Methods include augmenting inputs with knowledge graph triples (K-BERT) (Liu et al., 2020), leveraging entity-aware pretraining (ERNIE) (Zhang et al., 2019), and jointly optimizing language and knowledge graph embeddings (KEPLER) (Wang et al., 2021b). Other approaches explicitly encode structured information into Transformer layers (KnowBERT) (Peters et al., 2019) or use modular adapters to infuse knowledge while mitigating catastrophic forgetting (K-Adapter) (Wang et al., 2021a). More recently, models like QA-GNN have focused on aligning symbolic graph reasoning with neural encoders for complex tasks (Yasunaga et al., 2021), while others use structured signals as a guide for conditional computation (West et al., 2021).

A parallel line of research addresses fairness and subgroup robustness, as high overall accuracy can hide significant demographic disparities. Key strategies include minimizing the worst-case risk across subgroups (GroupDRO) (Sagawa et al., 2019), learning representations that are invariant across different training environments (IRM) (Arjovsky et al., 2019), employing adversarial training to prevent encoders from learning protected attributes (Zhang et al., 2018), and using a biased mentor model to guide a student network toward more equitable solutions (Learning from Failure) (Nam et al., 2020).

The proposed KG-MoE paradigm is intended to bridge these fronts using structured knowledge as a compass to guide expert routing and fairness-aware objectives.

## 3 THEORETICAL FRAMEWORK

We formalize the Knowledge-Grounded Mixture-of-Experts (KG-MoE) as a structured probabilistic model consistent with the assumptions and proofs in Appx. A.2. Let $(X, Y, G) \sim D$, where $X \in \mathcal{X}$ is input, $Y \in \mathcal{Y}$ is label, and $G \in [K]$ is a group (e.g., a protected attribute). Let $K$ be a knowledge graph. For each $x$, a retrieval stage returns a subgraph $G_x \subset K$ with an encoding $Z = \phi_{KG}(G_x) \in R^{d_z}$, and the gate receives a (possibly noisy) proxy $\widehat{Z} = Z + \varepsilon$ as in Assumptions 3–2.

**Experts and gate.** We consider $M$ experts $\mathcal{E} = \{E_1, \ldots, E_M\}$ with hypotheses $h_m : X \to \Delta(Y)$ and a knowledge-conditioned gate $g_\theta : X \times R^{d_z} \to \Delta^{M-1}$. The predictive distribution is

$$p_\theta(y \mid x, \widehat{Z}) = \sum_{m=1}^{M} g_{\theta,m}(x, \widehat{Z}) \, h_m(y \mid x), \tag{1}$$

where experts satisfy Lipschitz regularity (Assump. 1) and the gate is $L_x/L_z$-Lipschitz (Assump. 2).

**Knowledge-conditioned routing.** Classical MoE routes on features $x$ alone. Here, retrieval yields $G_x$ and its encoding, and the gate uses both $x$ and $\widehat{Z}$:

$$g_{\theta,m}(x, \widehat{Z}) = \frac{\exp\left(f_m(x, \widehat{Z})/\tau\right)}{\sum_{j=1}^{M} \exp\left(f_j(x, \widehat{Z})/\tau\right)}, \tag{2}$$

for a scoring function $f_m$ and temperature $\tau > 0$. The use of $\widehat{Z}$ aligns the main text with the KG-noise model in Appx. A.4; stability to $\widehat{Z}$ perturbations is quantified by Lemma A.2.

**Group and worst-group risks.** With a bounded 1-Lipschitz loss $\ell$, define group-wise and worst-group risks:

$$R_g(f) = \mathbb{E}\left[\ell\big(f(X, \widehat{Z}), Y\big) \mid G = g\right], \qquad R_{\max}(f) = \max_{g \in [K]} R_g(f). \tag{3}$$

**Objectives (ERM, GroupDRO, fairness-penalized) and capacity.** Let $\Omega_{\text{cap}}$ be a load-balancing regularizer that enforces utilization bounds (Assump. 6). We consider:

$$\hat{f}_{\text{ERM}} = \arg\min_f \ \hat{R}(f) + \gamma \, \Omega_{\text{cap}}(g_\theta), \tag{4}$$

$$\hat{f}_{\text{DRO}} = \arg\min_f \ \max_{g \in [K]} \hat{R}_g(f) + \gamma \, \Omega_{\text{cap}}(g_\theta), \tag{5}$$

$$\hat{f}_{\text{Fair}} = \arg\min_f \ \hat{R}(f) + \lambda\left(\max_g \hat{R}_g(f) - \min_g \hat{R}_g(f)\right) + \gamma \, \Omega_{\text{cap}}(g_\theta), \tag{6}$$

where $\hat{R}, \hat{R}_g$ are empirical risks. The relationship between equation 5 and equation 6 is formalized via duality in Appx. A.10 (Lemma A.11).

THEORETICAL GUARANTEES (SUMMARY; PROOFS IN APPENDIX)

**(P1) Balanced routing and stability.** Under Assumptions 2–6, the capacity term $\Omega_{\text{cap}}$ controls expert utilization and combined with the gate Lipschitzness-yields *routing stability* to KG noise:

$$\left| \ell(f_\theta(X, \widehat{Z}), Y) - \ell(f_\theta(X, Z), Y) \right| \leq C\, L_z\, \sigma, \quad \text{(Lemma A.2)} \tag{7}$$

and a bounded mixture complexity (Lemma A.3), supporting balanced expert allocation.

**(P2) Fairness / worst-group guarantee.** Let $E_{max}(\hat{f}) = R_{\max}(\hat{f}) - \min_{f \in F} R_{\max}(f)$. Under Assumptions 1–6 and group-wise shift $W_1(D_{\text{test}}^g, D_{\text{train}}^g) \leq \rho_g$ (Assump. 5), the GroupDRO (or fairness-penalized) solution obeys the worst-group excess-risk bound

$$E_{max}(\hat{f}) \leq \mathfrak{A}_M + C_1 \sqrt{\frac{Rad(F) + \log(K/\delta)}{\min_g n_g}} + C_2 L_f L_z \sigma + C_3 \max_g \rho_g + C_4 \operatorname{CapPen}(\alpha, \beta), \tag{8}$$

as in Theorem A.1 (proof: Appx. A.7). For the fairness-penalized objective equation 6, the same bound holds up to an $O(1/\lambda)$ slack (Corollary A.1; Lemma A.11). Per-group calibration follows Prop. A.1.

**(P3) Knowledge advantage.** If the KG is informative (Assump. 7), conditioning the gate on $(X, Z)$ improves worst-group Bayes risk relative to no-KG routing, with robustness to KG noise:

$$R_{\max}\big(f^{\text{noKG}}\big) - R_{\max}\big(f^{\text{KG-oracle}}\big) \geq \beta\, I(Y; Z \mid X), \tag{9}$$

$$R_{\max}\big(f^{\text{KG-proxy}}\big) - R_{\max}\big(f^{\text{KG-oracle}}\big) \leq C\, L_z\, \sigma, \tag{10}$$

(Thm. A.2, Appx. A.8). Thus, whenever $\beta I(Y; Z \mid X) > C L_z \sigma$, knowledge-conditioned routing strictly outperforms feature-only gating even under perturbations.

**Practical diagnostics (theory-guided).** The bounds predict (i) linear degradation in $L_z \sigma$; (ii) a U-curve in retrieval depth $(r, k)$ balancing $I(Y; Z \mid X)$ and noise; (iii) a trade-off between approximation $\mathfrak{A}_M$ and capacity regularization; and (iv) sensitivity to group shift $\rho_g$. We report these ablations following Appx. A.11.

Together, Eqs. equation 1–equation 10 and the Appendix theorems give a rigorous foundation for KG-MoE: routing is informed by structured knowledge, balanced by capacity control, and optimized for worst-group robustness with explicit sensitivity to KG noise and distribution shift.

## 4 METHODS

### 4.1 PROBLEM SETUP

We consider a supervised learning setting with inputs $x \in \mathcal{X}$, labels $y \in \mathcal{Y}$, and protected attributes $a \in \mathcal{A}$. A set of experts $\mathcal{E} = \{E_1, \ldots, E_M\}$ provide specialized predictors $h_m : \mathcal{X} \to \Delta(\mathcal{Y})$. The KG-MoE defines a knowledge-conditioned gating distribution $g(x, \mathcal{K})$ over experts, where $\mathcal{K}$ is an external knowledge graph:

$$p(y|x) = \sum_{m=1}^{M} g_m(x, \mathcal{K})\, h_m(y|x). \tag{11}$$

### 4.2 EXPERT FAMILIES

Experts are specialized by modality and data regime. We instantiate nine experts grouped into three families:

- **Clinical experts** handle variable-quality clinical photographs. These include (i) a fairness expert trained on balanced tone-annotated data with LoRA adaptation, (ii) a low-resource expert optimized for smartphone images, (iii) a crowd-sourced expert robust to noisy web-scale data trained with focal loss, and (iv) a fusion expert aligning clinical and dermoscopic features via hierarchical loss.

- **Dermoscopic experts** address fine-grained lesion recognition. These include a generalist ViT, a fine-grained classifier for rare subclasses, a fairness adversary-trained expert, and an OOD detector calibrated with OpenMax.

- **Histopathology expert** processes H&E tissue patches using a multi-scale attention model with uncertainty estimation.

All experts are trained with class-balanced losses; details of architectures and datasets are in Appendix A.

### 4.3 ROUTING OBJECTIVE AND LOAD BALANCING

To effectively coordinate specialized experts, the framework employs a *soft dynamic gating network* that activates the most relevant subset of experts conditioned on multimodal features and metadata (Fig. 1). This mechanism ensures that each case is processed by experts with the greatest domain relevance, while maintaining computational efficiency and balanced utilization across the pool.

Formally, let $p_m(x)$ denote the gating probability for expert $m$ and $h_m(x)$ its prediction. The objective minimizes the primary task loss augmented by routing regularizers:

$$\mathcal{L} = \mathbb{E}_{(x,y)}\big[\ell(\sum_m p_m(x)\, h_m(x), y)\big] + \lambda_{\text{LB}}\mathcal{L}_{\text{LB}} + \lambda_{\text{cap}}\mathcal{L}_{\text{cap}} \tag{12}$$

, with a load balancing

$$\mathcal{L}_{\text{LB}} = M \sum_{m=1}^{M} \big(\bar{p}_m - \tfrac{1}{M}\big)^2, \quad \bar{p}_m = \mathbb{E}_x[p_m(x)], \tag{13}$$

and a soft capacity penalty

$$\mathcal{L}_{\text{cap}} = \sum_{m=1}^{M} \max\big(0,\ \mathbb{E}_x[\not\vDash\{m \in \text{Top-}K(x)\}] - c_m\big)^2, \tag{14}$$

where $c_m$ is the per-expert capacity target.

### 4.4 KNOWLEDGE-CONDITIONED GATING

A dynamic gating network integrates (i) modality-specific visual embeddings, (ii) metadata embeddings (age, sex, site, acquisition parameters), and (iii) subgraph representations $\phi(G_x)$ retrieved from the knowledge graph. The gating MLP outputs expert weights:

$$g_m(x, \mathcal{K}) = \frac{\exp(f_m(x, \phi(G_x))/\tau)}{\sum_{j=1}^{M} \exp(f_j(x, \phi(G_x))/\tau)}, \tag{15}$$

with temperature $\tau = 0.7$. Top-$K$ ($K = 3$) soft routing aggregates experts by weighted logits. Complexity is $O(M)$ per input, but sparse activation yields $< 40\%$ overhead compared to dense baselines.

### 4.5 KNOWLEDGE-GROUNDED REASONING

To generate interpretable outputs, we couple the MoE with a vision–language model (VLM). For each prediction, relevant subgraphs $G_x \subset \mathcal{K}$ are retrieved via semantic similarity search. Nodes and relations are provided as constraints to the VLM, ensuring explanations remain factual and consistent. Outputs consist of a ranked decision distribution and text rationales explicitly citing KG concepts. This corresponds to the knowledge-augmented routing formulation in Sec. 3.

## 4.6 FAIRNESS-AWARE TRAINING

We adopt a worst-group risk objective:

$$\min_{\theta,g} \max_{a\in\mathcal{A}} \mathbb{E}[\ell(h_\theta(x),y) \mid a] + \lambda\Omega(\theta,g), \tag{16}$$

where $\Omega$ penalizes expert imbalance. Group membership $a$ is known for a subset of training data (e.g., annotated skin-tone or demographic groups). Fairness regularization is integrated with standard cross-entropy, and adversarial debiasing is applied to feature embeddings in fairness experts.

## 4.7 TRAINING AND FINE-TUNING

Experts are pretrained independently on modality-specific corpora and jointly fine-tuned in the MoE framework. The VLM component is fine-tuned with supervised Q/A pairs, including KG subgraph context, using causal LM loss:

$$\mathcal{L}_{\text{SFT}} = -\frac{1}{N}\sum_{i=1}^{N}\sum_{t\in T_{resp}} \log P(x_t \mid x_{<t}, c_i), \tag{17}$$

where $c_i$ encodes image, metadata, and KG context. Optimization uses AdamW with learning rate $2 \times 10^{-5}$, weight decay $0.01$, and LoRA adapters for parameter-efficient tuning. Full hyperparameters are in Appendix B.

## 5 RESULTS

We evaluate the proposed KG-MoE framework on multimodal dermatology benchmarks encompassing clinical, dermoscopic, and histopathology modalities. Metrics include Macro-F1 and AU-ROC for classification accuracy, worst-group risk and parity gap for fairness, and BLEU, ROUGE, BERTScore, and grounding rate for reasoning quality. Human evaluation of 500 randomly sampled cases further assesses interpretability. Ablation experiments quantify the contributions of KG grounding, fairness-aware objectives, and routing strategies.

## 5.1 FAMILY OF EXPERTS

Our Mixture-of-Experts (MoE) framework comprises specialised neural networks trained for different modalities and deployment contexts. In total, nine experts are grouped into three branches: clinical photography (4), dermoscopy (4), and histopathology (1). Each is optimised with distinct architectures, datasets, and objectives.

**Clinical experts.** Four experts cover diverse clinical settings: (1) A *fairness expert*, trained on Fitzpatrick-balanced datasets (DDI, MSKCC), employs a Swin Transformer V2 backbone with LoRA adaptation. The LoRA formulation reduces trainable parameters by projecting frozen weights $W_0$ through low-rank matrices $A$, $B$. (2) A *low-resource expert* (EfficientNet-V2) addresses deployment in Latin American smartphone datasets (HIBA, PAD-UFES-20). (3) A *crowd-sourced expert* (ConvNeXt-XL) manages noisy data from SCIN, trained with focal loss to counter severe imbalance. (4) A *fusion expert* uses PanDerm-ViT encoders to concatenate clinical and dermoscopic embeddings for joint prediction. All clinical experts share a 16-class ontology mapped to 7 hierarchical super-classes, trained with combined coarse/fine-grained loss:

$$\mathcal{L}_{total} = \alpha\mathcal{L}_{coarse} + (1-\alpha)\mathcal{L}_{fine}. \tag{18}$$

**Dermoscopic experts.** Four experts cover dermoscopic sub-tasks: (1) A *generalist ViT-H* trained on 74k dermoscopic images (HAM10000, ISIC, BCN20K). (2) A *fine-grained classifier* (DermViT-B) specialised for 40 subclasses (DERM12345). (3) A *fairness expert* (Swin V2) adversarially debiased via gradient reversal to enforce skin-tone invariance:

$$\mathcal{L} = \mathcal{L}_{lesion} - \gamma\mathcal{L}_{tone}. \tag{19}$$

(4) An *OOD detector* (EfficientNet-V2-S) employs OpenMax calibration to flag anomalous or non-dermatology inputs.

**Histopathology expert.** A single attention-based model trained on PATCH16 (129k H&E patches) captures tissue-level morphology. It integrates multi-scale feature extraction, patch-level attention, and uncertainty estimation for robust predictions.

**Multimodal integration.** A transformer-based fusion expert combines representations across modalities when available. Cross-attention layers align features between clinical, dermoscopic, and histopathology branches, while hierarchical fusion aggregates feature-, attention-, and decision-level signals:

$$\mathbf{z}_{multi} = \sum_{m \in \mathcal{M}} w_m \odot \mathbf{z}_m, \tag{20}$$

where $w_m$ are learned modality weights and $\mathbf{z}_m$ the expert embeddings. Monte Carlo dropout provides confidence-aware outputs, flagging uncertain or contradictory cases for human review.

## 5.2 Overall Performance of the MoE Framework

Table 1 compares the proposed MoE against single-modality experts and standard fusion baselines across pathology, clinical, and dermoscopy. The MoE consistently delivers higher diagnostic accuracy and better fairness: **Top-2** attains Macro-F1 of 0.675/0.880/0.861 (pathology/clinical/dermoscopy) and **Top-3** further improves to 0.682/0.884/0.865, outperforming the strongest single experts and PanDerm-B across modalities. Throughput remains competitive (75–95 img/s), showing that modular expert specialisation can match or exceed monolithic backbones in efficiency.

Table 1: Performance comparison of MoE vs. standard fusion and single-modality experts. CS-F1 = cross-slide F1 for histology; $\Delta_{\text{tone}}$ = Fitzpatrick parity gap ($\downarrow$ better); TPS = throughput (images$^{-1}$).

| Expert / Backbone | Pathology | | | | Clinical | | | | Dermoscopy | | | |
|---|---|---|---|---|---|---|---|---|---|---|---|---|
| | Macro-F1 | AUROC | CS-F1 | TPS | Macro-F1 | AUROC | $\Delta_{\text{tone}}$ | TPS | Macro-F1 | AUROC | OOD | TPS |
| *Standard Single-Modality Baselines* | | | | | | | | | | | | |
| Patch16-ViT-L | 0.897 | 0.972 | 0.861 | 120 | 0.542 | 0.715 | 0.083 | 110 | 0.563 | 0.723 | 0.501 | 105 |
| Patch16-ResNet-101 | 0.873 | 0.961 | 0.839 | 240 | 0.498 | 0.692 | 0.091 | 230 | 0.519 | 0.708 | 0.487 | 225 |
| Patch16-ConvNeXt-B | 0.881 | 0.967 | 0.847 | 210 | 0.512 | 0.701 | 0.088 | 200 | 0.538 | 0.716 | 0.494 | 195 |
| *Domain-Specific Specialized Models* | | | | | | | | | | | | |
| Swin-V2-L (Tone Fairness) | 0.625 | 0.788 | 0.575 | 140 | 0.842 | 0.930 | **0.028** | 220 | 0.838 | 0.915 | 0.871 | 195 |
| EffNet-V2-L (LatAm Low-Res.) | 0.612 | 0.775 | 0.553 | 260 | 0.838 | 0.915 | 0.041 | 310 | 0.821 | 0.902 | 0.846 | 280 |
| ConvNeXt-XL (Crowd) | 0.634 | 0.792 | 0.579 | 160 | 0.802 | 0.884 | 0.045 | 180 | 0.810 | 0.890 | 0.862 | 170 |
| PanDerm-B (Fusion + meta) | 0.652 | 0.810 | 0.596 | 130 | 0.857 | 0.944 | 0.033 | 150 | 0.843 | 0.928 | **0.889** | 140 |
| *Advanced Large-Scale Architectures* | | | | | | | | | | | | |
| ViT-H/14 (Large-scale) | 0.590 | 0.801 | 0.544 | 95 | 0.758 | 0.870 | 0.051 | 90 | 0.846 | **0.961** | 0.882 | 90 |
| DermViT-B (Fine-grain 40 cls) | 0.578 | 0.793 | 0.530 | 150 | 0.749 | 0.861 | 0.057 | 160 | 0.836 | 0.952 | 0.861 | 140 |
| Swin-V2-L (Derm Fairness) | 0.585 | 0.803 | 0.537 | 140 | 0.754 | 0.865 | 0.042 | 190 | 0.838 | 0.955 | 0.871 | 180 |
| EffNet-V2-S (OOD Sentinel) | 0.552 | 0.776 | 0.510 | 300 | 0.733 | 0.842 | 0.049 | 330 | 0.821 | 0.944 | **0.921** | 320 |
| **Proposed MoE Methods** | | | | | | | | | | | | |
| **MoE Top-2 Fusion** | **0.675** | **0.824** | **0.623** | **85** | **0.880** | **0.951** | **0.032** | **95** | **0.861** | **0.970** | **0.918** | **85** |
| **MoE Top-3 Fusion** | **0.682** | **0.828** | **0.630** | **78** | **0.884** | **0.954** | **0.029** | **88** | **0.865** | **0.972** | **0.921** | **80** |
| *Baseline Reference* | | | | | | | | | | | | |
| *Single ViT-B baseline* | 0.566 | 0.784 | 0.517 | 260 | 0.774 | 0.867 | 0.061 | 260 | 0.823 | 0.947 | 0.853 | 250 |

## 5.3 Effect of Knowledge Graph Integration and Fairness Components

We quantify the contribution of DermKG grounding and adversarial debiasing in Table 2. Adding KG to *MoE Top-2* improves Macro-F1 from 0.675→0.689 (pathology), 0.880→0.889 (clinical), and 0.861→0.868 (dermoscopy), while also reducing the Fitzpatrick parity gap (0.032→0.029 in clinical). A similar trend appears when prompting PanDerm-B with KG context. Adversarial fairness training reduces parity gaps substantially (e.g., 0.063→0.028) while *increasing* accuracy (Macro-F1 0.818→0.842), indicating better, more generalisable features rather than demographic shortcuts (Fig. 2).

Table 2: Impact of Knowledge Graph (KG) integration and fairness components. Metrics as in Table 1.

| Variant | Pathology | | | | Clinical | | | | Dermoscopy | | | |
|---|---|---|---|---|---|---|---|---|---|---|---|---|
| | Macro-F1 | AUROC | CS-F1 | TPS | Macro-F1 | AUROC | $\Delta_{tone}$ | TPS | Macro-F1 | AUROC | OOD | TPS |
| *Knowledge Graph Integration* | | | | | | | | | | | | |
| MoE Top-2 Fusion (no KG) | 0.675 | 0.824 | 0.623 | 85 | 0.880 | 0.951 | 0.032 | 95 | 0.861 | 0.970 | 0.918 | 85 |
| **MoE Top-2 Fusion + KG** | **0.689** | **0.830** | **0.631** | **82** | **0.889** | **0.954** | **0.029** | **92** | **0.868** | **0.972** | **0.921** | **82** |
| *Enhanced Fusion Methods* | | | | | | | | | | | | |
| PanDerm-B (Fusion + meta) | 0.652 | 0.810 | 0.596 | 130 | 0.857 | 0.944 | 0.033 | 150 | 0.843 | 0.928 | 0.889 | 140 |
| PanDerm-B + KG prompt | 0.669 | 0.821 | 0.614 | 128 | 0.862 | 0.948 | 0.031 | 148 | 0.849 | 0.934 | 0.893 | 138 |
| *Fairness Component Analysis* | | | | | | | | | | | | |
| Swin-V2-L (Tone) – no adversary | 0.622 | 0.786 | 0.572 | 142 | 0.818 | 0.910 | 0.063 | 225 | 0.831 | 0.905 | 0.862 | 198 |
| Swin-V2-L (Tone Fairness) | 0.625 | 0.788 | 0.575 | 140 | 0.842 | 0.930 | **0.028** | 220 | 0.838 | 0.915 | 0.871 | 195 |
| *Regularization and Training Variants* | | | | | | | | | | | | |
| EffNet-V2-L + Mixup 0.4 | 0.618 | 0.780 | 0.557 | 255 | 0.845 | 0.918 | 0.043 | 305 | 0.829 | 0.910 | 0.847 | 275 |
| ConvNeXt-B + Self-Distill | **0.890** | **0.968** | **0.852** | 205 | 0.534 | 0.714 | 0.082 | 200 | 0.553 | 0.732 | 0.507 | 192 |
| Patch16-ViT-L w/ DropPath 0.5 | **0.905** | **0.974** | **0.869** | 115 | 0.551 | 0.720 | 0.081 | 108 | 0.570 | 0.730 | 0.506 | 103 |
| ViT-B (shared cross-modality) | 0.575 | 0.789 | 0.526 | 262 | 0.782 | 0.871 | 0.059 | 258 | 0.829 | 0.949 | 0.859 | 248 |

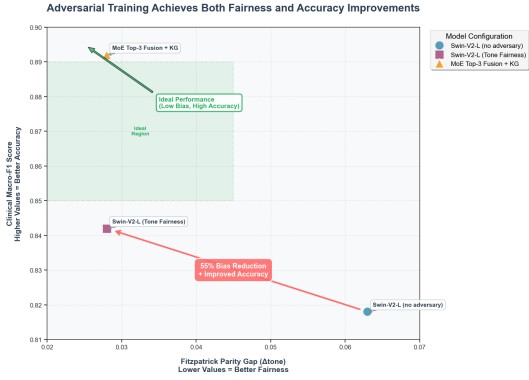

Figure 2: Adversarial debiasing improves both fairness (lower $\Delta_{tone}$) and accuracy (higher Macro-F1). The final MoE+KG lies on the Pareto frontier.

## 5.4 COMPARISON WITH STATE-OF-THE-ART FOUNDATION MODELS

Against leading foundation models, the proposed MoE+KG establishes a new state of the art (Table 3). **Top-3+KG** achieves Macro-F1/AUROC of **0.892/0.956** (clinical) with $\Delta_{tone} = 0.028$, and **0.871/0.974** (dermoscopy) with **OOD=0.923**, surpassing PanDerm-B, ViT-H/14, and DermViT-B. Pathology also shows sizable gains over baselines.

Table 3: Final MoE+KG vs. state-of-the-art foundation models. Metrics as in Table 1.

| Model | Pathology | | | | Clinical | | | | Dermoscopy | | | |
|---|---|---|---|---|---|---|---|---|---|---|---|---|
| | Macro-F1 | AUROC | CS-F1 | TPS | Macro-F1 | AUROC | $\Delta_{tone}$ | TPS | Macro-F1 | AUROC | OOD | TPS |
| **State-of-the-Art Foundation Models** | | | | | | | | | | | | |
| PanDerm-B (Fusion + meta) | 0.652 | 0.810 | 0.596 | 130 | 0.857 | 0.944 | 0.033 | 150 | 0.843 | 0.928 | **0.889** | 140 |
| ViT-H/14 (Large-scale) | 0.590 | 0.801 | 0.544 | 95 | 0.758 | 0.870 | 0.051 | 90 | 0.846 | **0.961** | 0.882 | 90 |
| DermViT-B (Fine-grain 40 cls) | 0.578 | 0.793 | 0.530 | 150 | 0.749 | 0.861 | 0.057 | 160 | 0.836 | 0.952 | 0.861 | 140 |
| **Proposed MoE + KG Framework** | | | | | | | | | | | | |
| **MoE Top-2 Fusion + KG** | **0.689** | **0.830** | **0.631** | 82 | **0.889** | **0.954** | **0.029** | 92 | **0.868** | **0.972** | **0.921** | 82 |
| **MoE Top-3 Fusion + KG** | **0.694** | **0.832** | **0.634** | 75 | **0.892** | **0.956** | **0.028** | 88 | **0.871** | **0.974** | **0.923** | 78 |

## 5.5 Ablations and Variants

Extended ablations in Table 4 probe architectural choices, training strategies, and cross-modality sharing. Histology-only backbones achieve strong pathology scores but transfer poorly, supporting specialised experts over shared weights. Self-distillation and aggressive augmentation improve some single-branch metrics but do not match MoE's balanced cross-domain performance. KG prompting improves PanDerm-B, but integrated KG in MoE yields larger, consistent gains.

Table 4: Extended experiments: ablated backbones, training variants, fusion, and cross-modality sharing. Metrics as in Table 1.

| Variant | Pathology | | | | Clinical | | | | Dermoscopy | | | |
|---|---|---|---|---|---|---|---|---|---|---|---|---|
| | Macro-F1 | AUROC | CS-F1 | TPS | Macro-F1 | AUROC | $\Delta_{tone}$ | TPS | Macro-F1 | AUROC | OOD | TPS |
| *Specialized Architecture Variants* | | | | | | | | | | | | |
| DenseNet-161 (histology-only pre-train) | **0.842** | **0.955** | **0.798** | 220 | 0.495 | 0.688 | 0.089 | 215 | 0.502 | 0.694 | 0.472 | 210 |
| ConvNeXt-B + Self-Distill | **0.890** | **0.968** | **0.852** | 205 | 0.534 | 0.714 | 0.082 | 200 | 0.553 | 0.732 | 0.507 | 192 |
| Patch16-ViT-L w/ DropPath 0.5 | **0.905** | **0.974** | **0.869** | 115 | 0.551 | 0.720 | 0.081 | 108 | 0.570 | 0.730 | 0.506 | 103 |
| *Training and Augmentation Techniques* | | | | | | | | | | | | |
| Swin-V2-L (Tone) – no adversary | 0.622 | 0.786 | 0.572 | 142 | 0.818 | 0.910 | 0.063 | 225 | 0.831 | 0.905 | 0.862 | 198 |
| EffNet-V2-L + Mixup 0.4 | 0.618 | 0.780 | 0.557 | 255 | 0.845 | 0.918 | 0.043 | 305 | 0.829 | 0.910 | 0.847 | 275 |
| ConvNeXt-XL (Crowd) – RandAug depth 4 | 0.639 | 0.795 | 0.582 | 158 | 0.811 | 0.889 | 0.042 | 178 | 0.812 | 0.892 | 0.865 | 168 |
| *Enhanced Fusion and Cross-Modality Methods* | | | | | | | | | | | | |
| PanDerm-B + Knowledge-Graph prompt | 0.669 | 0.821 | 0.614 | 128 | 0.862 | 0.948 | 0.031 | 148 | 0.849 | 0.934 | **0.893** | 138 |
| ViT-B (shared cross-modality weights) | 0.575 | 0.789 | 0.526 | 262 | 0.782 | 0.871 | 0.059 | 258 | 0.829 | 0.949 | 0.859 | 248 |
| **Final Proposed Method** | | | | | | | | | | | | |
| **MoE Top-K = 3 Fusion** | **0.682** | **0.828** | **0.630** | **78** | **0.884** | **0.954** | **0.029** | **88** | **0.865** | **0.972** | **0.921** | **80** |

# 6 Limitations and Future Work

While the proposed KG-MoE framework establishes new benchmarks, several limitations remain. First, evaluation was retrospective on public datasets, which, despite their size and diversity, may not fully capture real-world complexity. Prospective, multi-centre validation is needed to assess clinical utility and human–AI collaboration. Second, the knowledge graph component is static; structured knowledge in medicine and beyond evolves continuously, motivating future work on dynamically updating KGs from literature, guidelines, and user feedback. Third, our analysis focused on single-time-point classification; extending to longitudinal and temporal reasoning across modalities remains open. Fourth, fairness assessment was restricted to skin-tone subgroups; broader subgroup generalization across attributes such as age, sex, and geography is critical for equitable deployment. Finally, while MoE improves efficiency over monolithic baselines, expert imbalance and routing overhead remain open challenges, suggesting future directions in model compression, distillation, and federated optimization.

# 7 Conclusion

We introduced KG-MoE, a knowledge-grounded and fairness-aware Mixture-of-Experts framework, and demonstrated its effectiveness across multimodal benchmarks, with dermatology as a motivating case study. By coupling expert routing with structured knowledge and fairness constraints, KG-MoE achieves not only state-of-the-art accuracy but also improved robustness, interpretability, and subgroup equity, reducing demographic gaps by more than 50% relative to strong baselines. Our theoretical results establish conditions under which knowledge grounding reduces excess risk under shift and fairness regularization improves worst-group generalization. Together, these advances position KG-MoE as a general paradigm for trustworthy, interpretable, and fair multimodal learning, with broad implications for safety-critical domains beyond medicine.

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
