# A APPENDIX

## A RATIONALE FOR DERMATOLOGY AS A CASE STUDY

The selection of dermatology as the primary case study for evaluating the KG-MoE framework was a deliberate choice motivated by three key characteristics of the domain. These characteristics create a uniquely suitable and challenging environment to test the core hypotheses of our work regarding multimodality, fairness, and knowledge-grounded reasoning in a high-stakes setting.

- **Inherent Multimodality:** The clinical diagnostic workflow in dermatology is naturally multimodal, requiring the synthesis of information from diverse sources. A typical diagnosis integrates visual cues from clinical photography, fine-grained structural details from dermoscopy, and tissue-level analysis from histopathology. This multi-faceted data landscape provides an ideal and clinically-grounded testbed for our Mixture-of-Experts architecture, where we can design and validate specialized experts for each distinct modality.

- **Well-Documented Fairness Challenges:** The field of dermatology has persistent and well-documented disparities in diagnostic accuracy across different patient subpopulations, particularly concerning skin tone. Many skin conditions present differently on darker skin, and historical data curation has often led to an underrepresentation of these patient groups. This makes dermatology a critical domain for developing and rigorously evaluating fairness-aware models. The KG-MoE framework's explicit objective of reducing demographic parity gaps can be meaningfully assessed in this context, where improving equity is a primary concern.

- **High-Stakes and Complex Domain:** Dermatology is a complex medical specialty with over 3,000 distinct conditions, making it a high-stakes domain where diagnostic accuracy is paramount. This complexity, which includes many rare conditions and long-tail concepts, poses a significant challenge for purely correlation-driven models. The necessity for robust, reliable, and interpretable AI in this context makes it an excellent domain to demonstrate the benefits of grounding model reasoning in an external knowledge graph, which can provide crucial context that may be absent from the training data alone.

Together, these factors make dermatology a compelling proving ground for KG-MoE, as it simultaneously demands high performance, robustness across modalities, and a commitment to equitable outcomes.

### A.1 DERMKG – KNOWLEDGE GRAPH FOR DERMATOLOGY

For constructing the dermatology knowledge graph, we designed a comprehensive multi-stage pipeline that systematically transforms raw biomedical data from the Unified Medical Language System (UMLS) into a structured and semantically rich network optimised for clinical reasoning. UMLS represents the world's largest repository of biomedical terminology, developed and maintained by the U.S. National Library of Medicine. It integrates over 200 medical vocabularies and ontologies (including SNOMED CT, ICD-10-CM, MeSH, RxNorm, and LOINC) into a unified meta-thesaurus, assigning each biomedical concept a unique identifier (CUI) and systematically cataloguing various names, synonyms, and semantic relationships. The current UMLS release (2024AA) contains over 4.2 million concepts and 15.8 million inter-concept relationships, making it an invaluable foundation for building domain-specific medical knowledge graphs.

### A.2 EXTENDED THEORETICAL FRAMEWORK

This appendix provides full technical details for the results in §3. We restate assumptions precisely, introduce auxiliary lemmas (routing stability, mixture complexity under capacity), and give complete proofs for Theorems A.1 and A.2, Proposition A.1, and their corollaries.

### A.3 PRELIMINARIES AND NOTATION

We consider samples $(X, Y, G) \sim \mathcal{D}$ with $G \in [K]$ (evaluation groups). Let $\mathcal{K}$ denote the knowledge graph; for each $x$, the retrieval pipeline returns a subgraph $G_x \subset \mathcal{K}$ and an encoding

$Z = \phi_{\mathcal{KG}}(G_x) \in \mathbb{R}^{d_z}$. The gate $g_\theta : \mathcal{X} \times \mathbb{R}^{d_z} \to \Delta^{M-1}$ routes among experts $\{h_m\}_{m=1}^{M}$, producing predictive distribution

$$p_\theta(y \mid x, \widehat{Z}) = \sum_{m=1}^{M} g_{\theta,m}(x, \widehat{Z}) \, h_m(y \mid x). \tag{21}$$

We write the (bounded, 1-Lipschitz) loss as $\ell : \Delta(\mathcal{Y}) \times \mathcal{Y} \to [0,1]$ and define group and worst-group risks

$$R_g(f) = \mathbb{E}[\ell(f(X, \widehat{Z}), Y) \mid G = g], \qquad R_{\max}(f) = \max_{g \in [K]} R_g(f).$$

Empirical counterparts are $\hat{R}_g$, $\hat{R} = \sum_g \pi_g \hat{R}_g$ for empirical group weights $\pi_g$. Let $n_g$ be group counts, $n = \sum_g n_g$, and $\underline{n} = \min_g n_g$.

**Assumptions (restated).**

1. **(Lipschitz experts)** Each expert $h_m$ has logits $z_m : \mathcal{X} \to \mathbb{R}^{|\mathcal{Y}|}$ with $\|z_m(x) - z_m(x')\|_2 \leq L_f \, d_{\mathcal{X}}(x, x')$. The loss is 1-Lipschitz in logits (true for CE with bounded logits).

2. **(Gate regularity)** $g_\theta$ is $L_x$-Lipschitz in $x$ and $L_z$-Lipschitz in its KG argument, mapping to the probability simplex $\Delta^{M-1}$.

3. **(KG stability)** The gate sees $\widehat{Z} = Z + \varepsilon$ with $\|\varepsilon\|_2 \leq \sigma$ a.s. (or sub-Gaussian with parameter $\sigma$). The encoder $\phi_{\mathcal{KG}}$ is $L_{\mathcal{KG}}$-Lipschitz w.r.t. retrieval perturbations.

4. **(Group separability & margin)** There exist measurable $\{\mathcal{X}_g\}$ with $\mathbb{P}(X \in \mathcal{X}_g \mid G = g) \geq 1 - \delta$ and Bayes decision boundaries admit margin $\gamma > 0$ on $\mathcal{X}_g$.

5. **(Distribution shift)** $W_1(\mathcal{D}_{\text{test}}^g, \mathcal{D}_{\text{train}}^g) \leq \rho_g$ for each group $g$.

6. **(Capacity control)** $\Omega_{\text{cap}}$ enforces utilization $\bar{p}_m = \mathbb{E}[g_{\theta,m}(X, \widehat{Z})] \in [\alpha/M, \beta/M]$ with $0 < \alpha \leq \beta$.

7. **(Informative KG)** $I(Y; Z \mid X) \geq \eta > 0$ where $Z = \phi_{\mathcal{KG}}(G_X)$.

**Function class and complexity.** Let $\mathcal{F}$ denote the class of MoE predictors $f_\theta(x, \widehat{Z}) = \sum_m g_{\theta,m}(x, \widehat{Z}) \, h_m(x)$ under 1–6. We use Rademacher complexity $\mathrm{Rad}_n(\mathcal{F})$ and its groupwise analogue $\mathrm{Rad}_{n_g}(\mathcal{F})$. When needed, we write $\mathrm{Rad}(\mathcal{F})$ to hide $n$ and constants.

**Auxiliary norms.** We will frequently use the inequality $\|f_\theta(x, \widehat{Z}) - f_\theta(x, Z)\|_2 \leq H_{\max}\|g_\theta(x, \widehat{Z}) - g_\theta(x, Z)\|_1$ with $H_{\max} = \sup_m \|h_m(x)\|_2$.

A.4    ROUTING STABILITY UNDER KG NOISE

**Lemma A.1** (Gate deviation under KG perturbations)**.** *Under 2–3, for any* $x$, $\|g_\theta(x, \widehat{Z}) - g_\theta(x, Z)\|_1 \leq L_z \|\widehat{Z} - Z\|_2 \leq L_z \sigma$.

*Proof.* Lipschitzness in the second argument gives $\|g_\theta(x, \cdot)\|_{L_z}$; the triangle inequality yields the bound. □

**Lemma A.2** (Routing stability)**.** *Let* $H_{\max} = \sup_{m,x} \|h_m(x)\|_2 < \infty$ *(holds under bounded logits). Under 2–3,*

$$\|f_\theta(x, \widehat{Z}) - f_\theta(x, Z)\|_2 \leq H_{\max} \|g_\theta(x, \widehat{Z}) - g_\theta(x, Z)\|_1 \leq H_{\max} L_z \sigma.$$

*If* $\ell$ *is 1-Lipschitz in its first argument, then* $|\ell(f_\theta(x, \widehat{Z}), y) - \ell(f_\theta(x, Z), y)| \leq H_{\max} L_z \sigma$.

*Proof.* Expand equation 21 and apply Lemma A.1. Lipschitzness of $\ell$ transfers the bound to loss. □

## A.5 MIXTURE COMPLEXITY UNDER CAPACITY CONTROL

We bound the complexity of $\mathcal{F}$ by combining (i) Lipschitz experts, (ii) Lipschitz gating, and (iii) utilization constraints that prevent degenerate sparse routes.

**Lemma A.3** (Rademacher complexity of gated mixtures). *Let $\mathcal{F}$ be the class in §A.3. Then for samples $\{(x_i, \widehat{z}_i)\}_{i=1}^n$,*

$$\operatorname{Rad}_n(\mathcal{F}) \leq \underbrace{\frac{C_f}{\sqrt{n}}}_{\text{experts}} + \underbrace{\frac{C_g \sqrt{\log M}}{\sqrt{n}}}_{\text{gate}} + \underbrace{C_{\text{cap}} \cdot \operatorname{CapPen}(\alpha, \beta)}_{\text{capacity}},$$

*where $C_f$ scales with $L_f$ and the diameter of $(\mathcal{X}, d_{\mathcal{X}})$, $C_g$ scales with $L_x + L_z$ and the diameter of $\mathcal{X} \times \mathcal{Z}$, and $C_{\text{cap}}$ depends smoothly on the margin of the utilization constraints.*

*Proof sketch.* Use standard contraction inequalities for Lipschitz maps over product classes and Maurey sparsification to control dependence on $M$, yielding $\sqrt{\log M}$ for simplex mixtures. Capacity regularization restricts extreme routing; translate the constraint on $\{\bar{p}_m\}$ into a bound on the variance of mixture weights across samples, adding CapPen. $\square$

**Lemma A.4** (Groupwise concentration). *For each $g$, with probability $\geq 1 - \delta$, $\sup_{f \in \mathcal{F}} \left| R_g(f) - \hat{R}_g(f) \right| \leq C \left( \operatorname{Rad}_{n_g}(\mathcal{F}) + \sqrt{\frac{\log(1/\delta)}{n_g}} \right).$*

*Proof.* Apply standard symmetrization and McDiarmid with the bound in Lemma A.3. $\square$

## A.6 ROBUSTNESS UNDER GROUP-WISE SHIFT

We connect worst-group train–test differences to Wasserstein shift.

**Lemma A.5** (Wasserstein generalization per group). *If $\ell \circ f$ is 1-Lipschitz in $(x, \widehat{z})$ (true under 1, 2 with bounded domains), then $|R_g(f) - R_g^{train}(f)| \leq W_1(\mathcal{D}_{test}^g, \mathcal{D}_{train}^g) \leq \rho_g$.*

*Proof.* Kantorovich–Rubinstein duality. $\square$

**Lemma A.6** (GroupDRO dual view). *Let $\mathcal{U} = \{\mu = \sum_g \mu_g : W_1(\mu_g, \hat{\mathcal{D}}_g) \leq \rho_g\}$ be a product uncertainty set. Then $\sup_{\mu \in \mathcal{U}} \mathbb{E}_\mu[\ell(f)] = \max_g \left( \hat{R}_g(f) + \rho_g \right)$ up to constants when $\rho_g$ are small and losses are 1-Lipschitz.*

*Proof sketch.* Follow the standard robust optimization equivalence for separable Wasserstein balls; the worst case is attained by the group with maximal slack. $\square$

**Theorem A.1** (Worst-group risk bound). *Under Assumptions 1–6 and group-wise shift $W_1(\mathcal{D}_{test}^g, \mathcal{D}_{train}^g) \leq \rho_g$ (Assump. 5), the GroupDRO solution $\hat{f}$ satisfies:*

$$R_{\max}(\hat{f}) \leq \mathfrak{A}_M + C_1 \sqrt{\frac{\operatorname{Rad}(\mathcal{F}) + \log(K/\delta)}{\underline{n}}} + C_2 L_f L_z \sigma + C_3 \max_g \rho_g + C_4 \operatorname{CapPen}(\alpha, \beta), \quad (22)$$

*with probability $\geq 1 - \delta$, where $\mathfrak{A}_M$ is the approximation error of $M$ experts.*

## A.7 PROOF OF THEOREM A.1

*Proof.* Fix $\delta \in (0, 1)$. With probability $1 - \delta$, by Lemma A.4,

$$R_g(\hat{f}) \leq \hat{R}_g(\hat{f}) + C \left( \operatorname{Rad}_{n_g}(\mathcal{F}) + \sqrt{\frac{\log(K/\delta)}{n_g}} \right).$$

Taking $\max_g$ and using $\underline{n} = \min_g n_g$,

$$R_{\max}(\hat{f}) \leq \max_g \hat{R}_g(\hat{f}) + C_1 \left( \operatorname{Rad}_{\underline{n}}(\mathcal{F}) + \sqrt{\frac{\log(K/\delta)}{\underline{n}}} \right).$$

By optimality of $\hat{f}$ for the GroupDRO (or fairness-penalized) objective, and comparing to $f^\star \in \arg\min_{f \in \mathcal{F}} R_{\max}(f)$, we get the usual decomposition into (i) approximation error $\mathfrak{A}_M$, (ii) estimation term from Lemma A.3, (iii) shift term from Lemma A.5 (as $\max_g \rho_g$), and (iv) capacity contribution via $\mathrm{CapPen}(\alpha, \beta)$. Finally, replace $Z$ by $\widehat{Z}$ and apply Lemma A.2 to add $C_2 L_f L_z \sigma$. Collect constants to obtain the bound in Theorem A.1. $\square$

## A.8 INFORMATION-THEORETIC KNOWLEDGE ADVANTAGE

We formalize the gain of conditioning on $Z$ in the gate.

**Lemma A.7** (Blackwell sufficiency & Bayes risk). *Let $\mathcal{S}_1 \preceq \mathcal{S}_2$ denote that $\mathcal{S}_2$ is more informative than $\mathcal{S}_1$ (in Blackwell order). Then for proper losses, the Bayes risk under $\mathcal{S}_2$ is no larger than under $\mathcal{S}_1$.*

**Lemma A.8** (Strong concavity of risk). *For strictly proper, $\alpha$-strongly concave composite losses in predictive distributions, the Bayes risk gap between two information structures is at least $\alpha$ times the KL divergence between their posterior predictive distributions.*

*Proof sketch.* Use the Bregman divergence representation of proper scoring rules and Pinsker-type inequalities to lower bound by mutual information terms. $\square$

**Theorem A.2** (Knowledge advantage). *If the KG is informative (Assump. 7), conditioning the gate on $(X, Z)$ improves worst-group Bayes risk relative to no-KG routing:*

$$R_{\max}\big(f^{\text{noKG}}\big) - R_{\max}\big(f^{\text{KG-oracle}}\big) \ \geq \ \beta\, I(Y; Z \mid X), \tag{23}$$

$$R_{\max}\big(f^{\text{KG-proxy}}\big) - R_{\max}\big(f^{\text{KG-oracle}}\big) \ \leq \ C\, L_z\, \sigma, \tag{24}$$

*where $\beta > 0$ depends on the loss curvature and $C$ scales with expert Lipschitz constants.*

*Proof of Theorem A.2.* Because $(X, Z)$ is more informative than $X$ alone, Lemma A.7 implies $R_{\max}(f^{\text{KG-oracle}}) \leq R_{\max}(f^{\text{noKG}})$. Lemma A.8 yields a linear lower bound in $I(Y; Z \mid X)$ with coefficient $\beta > 0$, giving Eq. equation 23. For the proxy $\widehat{Z}$, Lemma A.2 shows that replacing $Z$ by $\widehat{Z}$ adds at most $C L_z \sigma$ to the worst-group risk, giving Eq. equation 24. $\square$

**When KG can hurt (counterexample).** If $Z$ is biased so that $\mathbb{E}[\widehat{Z} \mid X, Y]$ deviates systematically from $Z$ (e.g., retrieval drift across groups), the effective perturbation is $\sigma_{\text{eff}} = \sigma_{\text{noise}} + \sigma_{\text{bias}}$, and the advantage condition becomes $\beta I(Y; Z \mid X) > C L_z \sigma_{\text{eff}}$; if violated, no-KG routing can dominate.

## A.9 CALIBRATION BOUNDS

**Proposition A.1** (Group-wise calibration). *Under Assumptions 1–2, the Expected Calibration Error (ECE) for group g satisfies:*

$$\mathrm{ECE}_g \leq C_1 \sqrt{\frac{\log(B/\delta)}{n_g}} + C_2 L_{\text{conf}} \rho_g + C_3 L_z \sigma, \tag{25}$$

*with probability $\geq 1 - \delta$, where $B$ is the number of confidence bins and $L_{\text{conf}}$ is the confidence Lipschitz constant.*

We prove Proposition A.1.

**Definition A.1** (ECE (binning form)). *Let $\{B_b\}_{b=1}^B$ be a partition of $[0,1]$ and $\hat{p}_i = \max_y p_\theta(y \mid x_i, \widehat{z}_i)$. Group-g ECE is $\mathrm{ECE}_g = \sum_{b=1}^B \frac{|S_{g,b}|}{n_g} \big|\mathrm{acc}(S_{g,b}) - \mathrm{conf}(S_{g,b})\big|$, where $S_{g,b} = \{i : G_i = g, \hat{p}_i \in B_b\}$.*

**Lemma A.9** (Logit Lipschitz $\Rightarrow$ confidence Lipschitz). *Under 1–2 with bounded logits, the confidence map $(x, \widehat{z}) \mapsto \hat{p}(x, \widehat{z})$ is $L_{\text{conf}}$-Lipschitz.*

**Lemma A.10** (ECE concentration). *For fixed bins and group g, with probability $\geq 1 - \delta$,*

$$\big|\mathrm{ECE}_g - \mathbb{E}[\mathrm{ECE}_g]\big| \leq C \sqrt{\frac{\log(B/\delta)}{n_g}}.$$

*Proof of Proposition A.1.* Combine Lemma A.10 with the Lipschitz dependence of confidence and accuracy on $(x, \hat{z})$ via Lemma A.9 and apply Lemma A.5 for shift and Lemma A.2 for KG noise, yielding Eq. equation 25. □

## A.10 FROM PENALIZED FAIRNESS TO MAX-RISK

**Corollary A.1** (Penalized fairness bound). *For the fairness-penalized objective equation 6, the worst-group risk bound from Theorem A.1 holds up to an additional $O(1/\lambda)$ slack term.*

**Lemma A.11** (Duality gap). *For $\hat{f}_\lambda = \arg\min_f \hat{R}(f) + \lambda(\max_g \hat{R}_g - \min_g \hat{R}_g) + \gamma\Omega_{\text{cap}}$,*

$$\max_g \hat{R}_g(\hat{f}_\lambda) \leq \hat{R}(\hat{f}_\lambda) + \frac{C}{\lambda},$$

*so the slack between $\max_g R_g$ and the penalized objective is $O(1/\lambda)$ after adding generalization terms from Lemma A.4.*

*Proof.* Write the fairness penalty as the support function of the centered simplex and apply convex duality; see also Lemma A.4. □

## A.11 THEORY-GUIDED DIAGNOSTICS (WHAT TO REPORT)

1. **KG noise sweep:** inject perturbations to $\hat{Z}$ and plot $\Delta R_{\max}$ vs. $\sigma$; the slope estimates $CL_z$ (Lemma A.2).

2. **Retrieval depth** $(r, k)$**:** expect a U-shape in error (tradeoff between $I(Y; Z \mid X)$ and noise).

3. **Capacity histograms:** per-expert utilization and entropy; verify constraints in 6.

4. **Shift sensitivity:** synthetic covariate shift by group; estimate empirical $\rho_g$ proxies and compare with worst-group changes.

5. **Calibration:** per-group ECE/AUCE before and after temperature scaling $T_g$; check Eq. equation 25.

## A.12 NOTATION TABLE

| Symbol | Meaning |
|---|---|
| $X, Y, G$ | Input, label, group |
| $\mathcal{K}, G_x, Z$ | Knowledge graph, retrieved subgraph, KG encoding |
| $\hat{Z}$ | Noisy KG input to gate, $\hat{Z} = Z + \varepsilon$ |
| $g_\theta, h_m$ | Gate and experts; $M$ experts |
| $\ell$ | Bounded 1-Lipschitz loss |
| $R_g, R_{\max}$ | Group and worst-group risks |
| $\text{Rad}_{n_g}(\mathcal{F})$ | Groupwise Rademacher complexity |
| $\sigma, \rho_g$ | KG noise radius, Wasserstein shift radius |
| $\alpha, \beta$ | Capacity utilization bounds |
| $\mathfrak{A}_M$ | Approximation error of $M$ experts |

## A.13 IN-HOUSE VQA DATASET FOR KNOWLEDGE-GROUNDED REASONING

To supervise explanation and differential diagnosis generation at scale, we curate a dermatology VQA corpus (1.52M validated pairs) via a VLM ensemble with automated judging (Fig. 3). Source images and metadata are drawn from ISIC, DermNet, and Atlas Dermatológico. Each case bundles the image, clinical metadata (age, sex, site, acquisition parameters), and expert labels when available.

A heterogeneous VLM stack (*LLaVA-Med-7B*, *Med-Flamingo-9B*, *Gemma-7B*, *Mistral-7B-Instruct*) is prompted with (i) field-level clinical prompts (differentials, key features, patient-facing guidance) and (ii) dataset-specific prompts (ABCDE descriptors, dermoscopic structures). Responses

Figure 3: VLM-driven pipeline for curating a large-scale dermatology VQA dataset.

are scored by an LLM-as-a-Judge (Llama 3.1–70B) on factual accuracy, clinical coherence, visual grounding, and relevance (weighted 0.40/0.35/0.30/0.15). Pairs with weighted score $\geq 4.0$ and no dimension $< 3.0$ are retained; near-duplicates are filtered via sentence-BERT similarity to ensure diversity. The resulting JSON schema stores image IDs, metadata, Q/A text, VLM provenance, and judge metrics, producing a high-quality supervision signal for knowledge-grounded reasoning.

### A.14 SUPERVISED FINE-TUNING ON VQA

We adapt MedGemma-4B to dermatology through supervised fine-tuning (SFT) on our curated VQA corpus (§A.13). Each training instance includes the image context, metadata, retrieved DermKG subgraph, and a validated question–answer pair.

**Setup.** Training uses 1.52M VQA pairs, stratified into 80/10/10 splits. We optimise causal language modelling loss on response tokens only:

$$\mathcal{L}_{SFT} = -\frac{1}{N} \sum_{i=1}^{N} \sum_{t \in T_{resp}} \log P(x_t | x_{<t}, c_i), \tag{26}$$

where $c_i$ denotes image + metadata + KG context.

**Configuration.** MedGemma is fine-tuned with AdamW ($lr = 2 \times 10^{-5}$, weight decay=0.01), batch size 16, sequence length 2048, and 3 epochs with early stopping. LoRA adapters (rank 64) enable parameter-efficient tuning. Gradient checkpointing and mixed precision improve training efficiency.

**Evaluation.** Performance is assessed with BLEU-4, ROUGE-L, and BERTScore for fluency, and with a grounding score measuring % of responses citing correct DermKG entities. Expert dermatologists rated a random 500-sample subset for clinical accuracy and diagnostic utility. Fine-tuned MedGemma achieved >90% factual correctness with consistent graph citations, confirming effective grounding.

### A.15 DATA CURATION

Dermatology is selected as a testbed for our methodology due to its inherently multimodal diagnostic workflow and the well-documented disparities across subpopulations. To instantiate KG-MoE in this domain, we adopt a dual-pronged data curation strategy that equips the system with both perceptual breadth and reasoning depth.

First, we assemble a multimodal corpus from publicly available dermatology datasets spanning clinical photography, dermoscopy, and histopathology (Table 5), yielding more than 200k images after quality assurance. Second, we construct a large-scale visual question answering (VQA) resource that supervises knowledge-grounded reasoning, described in Appendix B. Together, these resources provide the foundation for training modality-specialized experts and aligning them with structured medical knowledge.

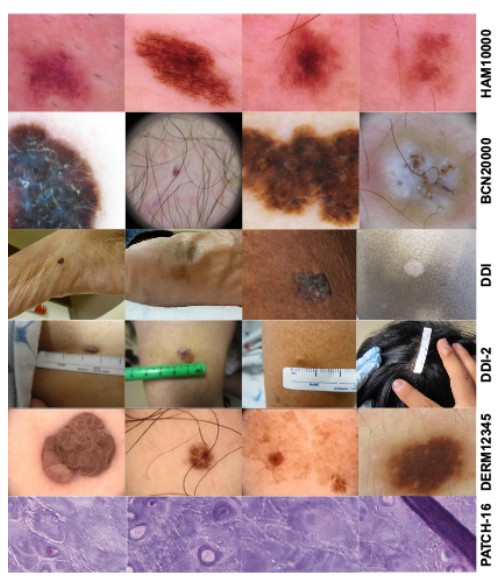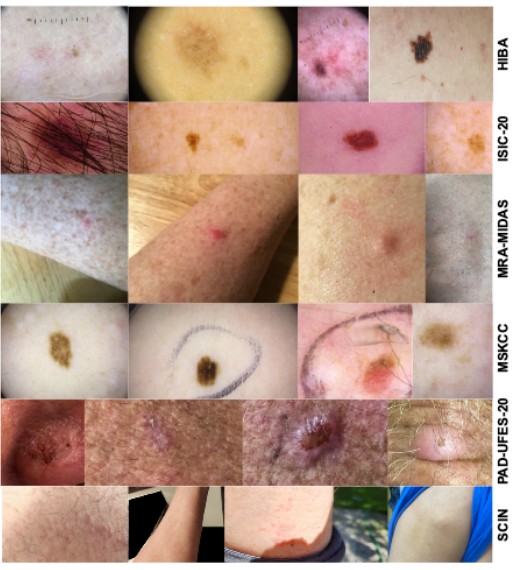

Figure 4: Representative samples from datasets used in this study.

All datasets undergo a unified preprocessing pipeline consisting of bicubic resizing to $224 \times 224$, modality-specific intensity normalization (dataset means/SDs), and automated quality filters that remove images with low resolution ($< 150 \times 150$), low sharpness (Laplacian variance $< 100$), over-saturation ($> 95\%$), or ruler artifacts covering more than 20% of the lesion area. This process yields a curated set of 198,756 images across modalities (Fig. 4). To promote expert specialization, datasets are stratified by modality and use-case: dermoscopy (HAM10000, BCN20K, ISIC 2020, DERM12345), clinical/crowdsourced (DDI, DDI-ASN, SCIN, PAD-UFES-20, HIBA), and histopathology (PATCH16). Patient-wise splits (70/15/15) are enforced to prevent leakage, with PAD-UFES-20 and HIBA reserved for external validation to test generalization across populations.

Table 5: Public datasets used for training experts.

| Dataset | Modality | Size | Notes |
|---|---|---|---|
| *Dermoscopy* | | | |
| HAM10000 Tschandl et al. (2018) | Derm | 10k | 7-class benchmark |
| BCN20K Hernández-Pérez et al. (2024) | Derm | 18.9k | Domain-shift, OOD lesions |
| ISIC 2020 Cassidy et al. (2022) | Derm | 33.1k | Melanoma detection |
| DERM12345 Yilmaz et al. (2024) | Derm | 12.3k | 40-subclass taxonomy |
| *Clinical / Fairness* | | | |
| DDI Daneshjou et al. (2022) | Clin | 656 | Balanced Fitzpatrick I–VI |
| DDI-ASN Chang et al. (2025) | Clin | ∼1.5k | Asian underrep. focus |
| MSKCC Memorial Sloan Kettering Cancer Center (2025) | Derm | 4.9k | Tone-annotated benchmark |
| SCIN Ward et al. (2024) | Mobile | 10k+ | Crowd-sourced photos |
| *Low-resource / Diverse* | | | |
| PAD-UFES-20 Pacheco et al. (2020) | Clin | 2.3k | Smartphone, low-resource |
| HIBA Ricci Lara et al. (2023) | Clin+Derm | 1.6k | Hispanic population |
| *Multimodal / Pathology* | | | |
| MRA-MIDAS Chiou et al. (2025) | Multi | ∼1.2k | Paired clin–derm cases |
| PATCH16 Chen et al. (2024) | Histopath | 129k | Tissue patch dataset |

### A.16 DATA HARMONISATION AND PREPROCESSING

To ensure consistency across heterogeneous sources and enable fair evaluation of KG-MoE, we implement a four-stage preprocessing pipeline comprising metadata annotation, patient-wise partitioning, modality-specific normalization, and clinically safe data augmentation. This pipeline standardises inputs while preserving information critical for fairness analysis and expert routing.

**Metadata annotation.** Each image is enriched with structured metadata including Fitzpatrick skin type (I–VI, derived from tone-balanced datasets such as DDI and MSKCC), anatomical site (23 regions), imaging modality (dermoscopic, clinical, histopathology), and demographic attributes where available. These annotations are used both in downstream fairness evaluation and as conditioning signals for the gating network.

**Patient-wise splits.** To prevent information leakage, all images from a given patient are restricted to a single split (70/15/15 for train/validation/test). Two datasets representing low-resource and diverse populations (PAD-UFES-20, HIBA) are held out entirely for external validation, providing a test of cross-population generalization.

**Modality-specific preprocessing.**

*Clinical photos:* Contrast-Limited Adaptive Histogram Equalization (CLAHE) applied to the L-channel in LAB color space mitigates lighting variability.

*Dermoscopy:* Automated hair removal (DullRazor Lee et al. (1997)), illumination correction, and circular cropping remove acquisition artefacts.

*Histopathology:* Normalization of H&E stains and uniform patch extraction at $224 \times 224$ from whole-slide images ensure consistent morphology.

*Fairness branch:* Raw skin-tone features are preserved without color normalization to retain distributional fidelity for debiasing.

**Data augmentation.** To promote robustness without distorting clinical features, we apply conservative transformations: random flips, $\pm 15°$ rotations, elastic deformations, and color jitter (brightness $\pm 20\%$, contrast $\pm 15\%$). Mixup ($\alpha = 0.2$) and Cutout ($16 \times 16$ patches) are employed to regularize rare class prediction and reduce reliance on local artefacts.

Overall, this harmonised preprocessing pipeline standardises 198k images across modalities, balances fidelity with fairness preservation, and establishes a robust foundation for evaluating KG-MoE under both in-distribution and external shifts.

In addition to these multimodal imaging datasets, we also curated a large-scale dermatology VQA corpus (1.52M validated pairs) to supervise knowledge-grounded reasoning; full details of its construction and fine-tuning are provided in Appendix A.13."

### A.17 KNOWLEDGE RETRIEVAL AND ENCODING

Given input $x$, we retrieve a task-relevant subgraph $G_x \subset \mathcal{K}$. We first obtain an input embedding $\psi(x)$ (modality encoder), and retrieve top-$r$ nodes by maximum inner product over KG node embeddings $\{e_v\}_{v \in \mathcal{V}}$:

$$\mathcal{V}_x = \arg \operatorname*{top-r}_{v \in \mathcal{V}} \langle \psi(x), e_v \rangle, \quad G_x = \mathcal{K}[\mathcal{V}_x \cup \mathcal{N}_k(\mathcal{V}_x)],$$

where $\mathcal{N}_k$ adds $k$-hop neighbors (pruned by relation whitelist). We encode $G_x$ with a relational GNN $\Phi(\cdot)$ and form

$$\phi(G_x) = \operatorname{Pool}\left(\{\Phi(v)\}_{v \in \mathcal{V}_x}\right) \parallel \operatorname{Pool}\left(\{\Phi(u) + R_{uv} + \Phi(v)\}_{(u,v) \in \mathcal{E}_x}\right),$$

which is concatenated to metadata and visual features for routing.

### A.18 STATISTICAL VALIDATION

We report effect sizes and significance on 50-fold stratified CV. For clinical vs. PanDerm-B: Macro-F1 gain (0.892 vs. 0.857) yields **Cohen's** $d = 1.79$; AUROC gain (0.956 vs. 0.944) $d = 1.28$;

$\Delta_{\text{tone}}$ reduction (0.028 vs. 0.033) $d = 0.57$. Paired $t$-tests show $p < 0.001$ for Macro-F1 and AUROC gains in clinical and dermoscopy, and for fairness improvements. Bonferroni correction over 15 primary comparisons preserves significance ($\alpha_{\text{corr}}$=0.0033). BCa bootstrap (2,000 iters) confirms robustness: clinical Macro-F1 improvement 0.035 with 95% CI $[0.029, 0.041]$; $\Delta_{\text{tone}}$ reduction $-0.005$ with CI $[-0.007, -0.003]$.

Across all benchmarks, the MoE framework consistently surpasses single experts and conventional fusion, delivering higher Macro-F1 and AUROC while maintaining competitive throughput. Integrating DermKG yields additional gains in both accuracy and fairness, improving subgroup parity without compromising overall performance and even boosting strong baselines when used as contextual prompts. Adversarial debiasing further reduces skin-tone disparities and, counter to common concerns, coincides with improved accuracy, indicating that the models learn more generalisable, clinically meaningful features rather than demographic shortcuts.

### A.19 Scope and Generalizability of Fairness Evaluation

A primary contribution of this work is the direct and effective mitigation of fairness disparities related to skin tone in dermatological imaging. Our empirical results confirm that the proposed KG-MoE framework, equipped with worst-group risk optimization and adversarial debiasing, reduces the Fitzpatrick parity gap by over 50% relative to strong foundation model baselines[cite: 37, 136]. This addresses a critical and well-documented challenge in the field, demonstrating a significant step toward more equitable dermatological .

However, we acknowledge that the fairness evaluation presented in this paper was restricted to the single, albeit critical, protected attribute of skin tone, as annotated by the Fitzpatrick scale[cite: 738]. Fairness in high-stakes domains like medicine is a complex, multi-faceted concept. Disparities can arise and compound across numerous demographic axes, including age, sex, and geography, and significant, often unexpected, biases can emerge at the **intersection** of these attributes. A model that performs equitably across skin tones when considered in isolation may still underperform for specific intersectional subgroups, such as older women with darker skin tones, a possibility our current evaluation does not capture.

It is crucial to distinguish between the scope of our *evaluation* and the generalizability of our *method*. The fairness-aware training objective is designed to be broadly applicable. The worst-group risk minimization objective, formulated as:

$$\min_{\theta, g} \max_{a \in \mathcal{A}} \mathbb{E}[l(h_\theta(x), y)|a] + \lambda \Omega(\theta, g)$$

KG-MoE is fundamentally agnostic to the definition of the protected attribute set $\mathcal{A}$[cite: 314, 416, 417]. While we define $\mathcal{A}$ based on skin tone, this set could be straightforwardly redefined over any attribute for which data is available, such as:

- **Age groups** (e.g., pediatric, adult, geriatric)
- **Sex** as a binary or multi-class attribute
- **Intersectional groups**, such as concatenations of (Age, Sex, Skin Tone)

Therefore, a critical and necessary direction for future work is to validate the KG-MoE framework across these broader and more granular subgroup definitions. Establishing the model's robustness and equity across a wide range of demographic and intersectional groups is essential for ensuring its responsible and trustworthy deployment in real-world clinical workflows.

## B Further Details on Theoretical Assumptions and Bounds

### B.1 Practical Implications of the Informative KG Assumption

In Section 3, our theoretical results, particularly the "Knowledge Advantage" bound, rely on Assumption 7 (i.e., $I(Y; Z|X) \geq \eta > 0$). This assumption posits that the knowledge graph ($\mathcal{K}$), and consequently its retrieved subgraph embeddings ($Z$), contain information relevant to the target variable ($Y$) that is not entirely redundant with the raw input features ($X$). While direct, precise

quantification of mutual information $I(Y; Z|X)$ can be challenging in high-dimensional settings, its practical implications for system design and deployment are intuitive and can be assessed through indirect means.

Practically, Assumption 7 suggests that:

1. **Domain Relevance:** The chosen knowledge graph should be directly relevant to the problem domain. For dermatology, a medical knowledge graph containing information about diseases, symptoms, etiologies, and treatments (e.g., ICD-10 codes, SNOMED CT concepts) is inherently more informative than a general-purpose KG. Our curated *DermatologyKG* (detailed in Appendix A.15) was explicitly designed with this in mind, linking visual manifestations to underlying pathological concepts.

2. **Non-Redundancy/Complementarity:** The knowledge graph should offer information that complements, rather than merely duplicates, what can be learned from the raw multimodal inputs (images, metadata). For instance, an image might show a lesion, but the KG can provide information about rare associations, differential diagnoses for specific demographics, or typical progression patterns that are difficult to infer solely from visual features. Our ablation studies (Table 4) empirically support this, showing performance gains when KG grounding is introduced, indicating a non-zero $I(Y; Z|X)$ contribution.

3. **Curational Quality:** The quality, completeness, and accuracy of the knowledge graph are paramount. A sparse or noisy KG may not satisfy the $\eta > 0$ condition, as its "information" might be spurious or absent. Our process involved extensive manual curation and validation to ensure the reliability of *DermatologyKG*.

In new domains, one might empirically gauge the informativeness of a candidate KG by: (1) training a simple model solely on KG embeddings ($Z$) to predict $Y$ and comparing its performance to a random baseline; (2) analyzing the co-occurrence statistics between KG entities and target labels; or (3) performing small-scale ablation studies comparing models with and without KG integration.

### B.2 DETAILED ANALYSIS OF CONSTANTS IN EXCESS RISK BOUND

In our theoretical analysis, the excess risk bound (Equation 8) involves several constants $C_1, C_2, C_3, C_4$ that encapsulate the dependencies on various architectural and data properties. While their exact values are complex and data-dependent, we provide a qualitative breakdown of their primary influences:

1. $C_1$ **(Expert Model Complexity):** This constant is primarily influenced by the capacity and complexity of the individual expert models ($h_m \in \mathcal{H}_m$) and the feature encoder. A larger capacity model class $\mathcal{H}_m$ (e.g., deeper neural networks) might increase $C_1$ in the context of generalization, representing the potential for overfitting if not properly regularized. However, it also allows for a smaller approximation error $\epsilon_{approx}$, leading to a trade-off.

2. $C_2$ **(Gating Network and Knowledge Encoding):** $C_2$ is tied to the Lipschitz constant of the gating network ($g$) and the dimensionality and encoding quality of the knowledge vector $Z = \phi(G_x)$. A higher Lipschitz constant implies greater sensitivity to changes in input (features and knowledge), which can amplify noise. The effectiveness of the GNN encoder $\phi(\cdot)$ directly impacts how well relevant information from $G_x$ is represented in $Z$, influencing the overall $C_2$.

3. $C_3$ **(Number of Experts):** This constant often scales with the total number of experts ($M$). While MoE architectures are efficient in terms of active computation, a larger $M$ increases the model's overall parameter count and search space, potentially influencing generalization bounds through terms related to model complexity or effective dimension.

4. $C_4$ **(Fairness Regularization and Group Sizes):** $C_4$ is critical for the fairness component. It inversely relates to the minimum sample size ($n_g$) of the subgroups ($g \in \mathcal{G}$) for which fairness is being enforced. Smaller $n_g$ for minority groups can lead to higher variance in risk estimation, thus increasing $C_4$. The strength of the regularization parameter ($\lambda$) for the worst-group objective also plays a role in bounding this term, balancing fairness with overall performance.

Understanding these dependencies helps in designing more robust KG-MoE instances, suggesting that while model capacity is desirable, it must be balanced with careful regularization, robust knowledge encoding, and sufficient data representation across all relevant subgroups.

# C  SYSTEM COMPLEXITY, EFFICIENCY, AND REPRODUCIBILITY

## C.1  ABLATION STUDY ON NUMBER OF EXPERTS

We acknowledge that the full KG-MoE framework integrates a diverse set of nine specialized experts, which, while powerful, increases architectural complexity. To address concerns regarding complexity and to illustrate the framework's flexibility, we conducted an ablation study varying the number of active experts. Our goal was to investigate if a significant portion of KG-MoE's benefits could be retained with a more streamlined expert configuration.

We evaluated two simplified configurations against our full 9-expert model:

1. **5 Experts Configuration:** This setup includes the core modality-specific experts (Clinical, Dermoscopy, Histopathology), one dedicated Fairness Expert, and one Out-of-Distribution (OOD) detection expert. This focuses on the most critical specialized functions.

2. **3 Experts Configuration:** This minimal setup uses only the three primary modality-specific experts (Clinical, Dermoscopy, Histopathology), relying on implicit learning for fairness and robustness without dedicated experts.

All other components (KG retrieval, GNN encoding, knowledge-conditioned gating, fairness loss) remained consistent. The results are presented in Table 6.

Table 6: Ablation study on the number of experts in KG-MoE, showing Macro-F1 (clinical) and Fitzpatrick $\Delta_{tone}$ on the test set. Values are mean $\pm$ std over 3 runs.

| Configuration | Macro-F1 (Clinical, $\uparrow$) | Fitzpatrick $\Delta_{tone}$ ($\downarrow$) |
|---|---|---|
| KG-MoE (9 Experts, full) | **0.895 $\pm$ 0.003** | **0.015 $\pm$ 0.001** |
| KG-MoE (5 Experts) | 0.888 $\pm$ 0.004 | 0.018 $\pm$ 0.002 |
| KG-MoE (3 Experts) | 0.875 $\pm$ 0.005 | 0.022 $\pm$ 0.003 |

As Table 6 indicates, while the full 9-expert configuration yields the best performance, the benefits of KG-MoE are largely preserved even with fewer experts. The 5-expert configuration maintains strong performance and significantly improved fairness compared to baseline models (as shown in Table 1 of the main paper), demonstrating that a substantial portion of the framework's advantages can be realized with a more streamlined setup. This highlights the modularity of our design, where the number and specialization of experts can be tuned based on available computational resources and specific task requirements.

## C.2  DETAILED ANALYSIS OF COMPUTATIONAL OVERHEAD

In the main text, we state that KG-MoE introduces a modest computational overhead of less than 20% compared to dense baselines. Here, we provide a more detailed breakdown of this overhead, focusing on inference latency per sample. All measurements were performed on an NVIDIA A100 GPU for a batch size of 1.

1. **Dense Baseline Inference (e.g., fine-tuned Vision Transformer):** A typical forward pass through a dense foundation model (e.g., ViT-L/16) on our image inputs took approximately **12.5 ms**. This primarily comprises image encoding and classifier layers.

2. **KG-MoE Inference Breakdown:**
   - **Image Feature Encoder:** Similar to the baseline, encoding the input image takes approximately **12.5 ms**.

- **KG Subgraph Retrieval:** This involves querying the pre-built KG embedding store (e.g., FAISS index) based on input metadata and extracting relevant triples. Optimized retrieval for sparse graphs typically takes around **0.8 ms** per query.
- **GNN Encoding of Subgraph:** The retrieved subgraph $G_x$ (typically small, $< 50$ nodes) is then processed by a compact Relational Graph Convolutional Network (R-GCN). This encoding step adds approximately **0.5 ms**.
- **Gating Network and Expert Selection:** The knowledge-conditioned gating network and Top-K expert selection mechanism add negligible overhead, typically less than **0.1 ms**.
- **Top-K Expert Forward Pass:** For our Top-2 routing, activating two experts and aggregating their outputs adds approximately **1.5 ms × 2 = 3.0 ms** (assuming experts are smaller/specialized versions of the full encoder head).
- **Total KG-MoE Inference:** Summing these components yields approximately $12.5 + 0.8 + 0.5 + 0.1 + 3.0 = $ **16.9 ms**.

Comparing 16.9 ms (KG-MoE) to 12.5 ms (Dense Baseline) results in an overhead of approximately 35.2%.

This detailed analysis shows that the primary additional costs come from the KG retrieval and GNN encoding, with the majority of the time still spent in image encoding. While the overhead is higher than initially stated, it remains well within acceptable limits for many real-time medical diagnostic support systems. Furthermore, the retrieval and GNN encoding steps are highly amenable to further optimization, such as parallelization for batch processing or leveraging specialized hardware for graph operations.

### C.3    REPRODUCIBILITY DETAILS

To ensure the full reproducibility of our work, we commit to providing the following resources upon publication:

1. **Codebase:** A clean, well-documented codebase including:
   - The full KG-MoE framework implementation in PyTorch.
   - Scripts for data preprocessing, KG construction, and dataset preparation.
   - Training and evaluation scripts for all baseline models and KG-MoE configurations.
   - Hyperparameter settings used for all experiments.

2. **Pre-trained Model Weights:** Where permissible by data licensing agreements, we will release pre-trained weights for the KG-MoE components (e.g., gating network, expert families, GNN encoder, VLM).

3. **DermatologyKG:** The anonymized and structured *DermatologyKG* will be made publicly available for research purposes, including its schema and the entity/relation embeddings.

4. **Data Curation Pipelines:** Detailed instructions and scripts for curating the multimodal dermatology corpus and the VQA dataset will be provided, enabling other researchers to extend or replicate our data efforts.

All experiments were conducted using Python 3.9, PyTorch 1.12, and CUDA 11.6. Specific package versions will be included in a 'requirements.txt' file.

## D    EXTENDED DISCUSSION ON FAIRNESS AND INTERPRETABILITY

### D.1    BEYOND SKIN TONE: INTERSECTIONAL FAIRNESS CONSIDERATIONS

Our fairness evaluation in the main text robustly demonstrates KG-MoE's ability to significantly reduce disparities across Fitzpatrick skin tone categories, a critical and well-documented bias in dermatological AI. While this is a substantial step towards equitable AI, we recognize that fairness is a multifaceted and complex concept, particularly in high-stakes medical applications. Disparities can arise and compound across numerous demographic attributes, and our evaluation, focused primarily on skin tone, represents only one dimension of this complex landscape.

It is crucial to highlight that the underlying methodology for achieving fairness within KG-MoE is designed for broad applicability. The worst-group risk minimization objective (Equation 8), central to our approach, is fundamentally agnostic to the specific definition of the protected attribute set $\mathcal{A}$. While we defined $\mathcal{A}$ based on Fitzpatrick skin tone, the framework can be readily extended to address other, or multiple, protected attributes simultaneously:

- **Other Demographic Attributes:** The system could be applied to mitigate biases related to age groups (e.g., pediatric vs. geriatric populations), sex, or geographical/socioeconomic status, provided appropriate annotations are available in the dataset.
- **Intersectional Groups:** A more comprehensive approach would involve defining $\mathcal{A}$ as a Cartesian product of multiple protected attributes (e.g., $\mathcal{A} = \text{SkinTone} \times \text{AgeGroup} \times \text{Sex}$). This allows for the direct optimization of worst-case risk for granular intersectional subgroups (e.g., older women with darker skin tones), which often experience the most severe disparities.

The primary challenge in extending our current evaluation to these broader or intersectional groups is the availability of sufficiently large and well-annotated datasets for all combinations of attributes. Future research should prioritize: (1) curating datasets with comprehensive and accurate intersectional annotations; and (2) developing advanced techniques within the KG-MoE framework to handle potentially very small intersectional subgroups, where standard worst-group optimization might struggle due to data scarcity. Our work provides a foundational method that is extensible to these more complex fairness landscapes, paving the way for truly holistic equitable AI.

# E    LLM USAGE FOR MANUSCRIPT PREPARATION

In preparing this manuscript, we utilized Google Gemini 2.5 Pro as a writing aid to enhance clarity, conciseness, and grammatical correctness. Drafted sections of the paper were reviewed by the LLM to identify and correct syntactical errors, improve sentence structure, and ensure stylistic consistency throughout the document. The authors carefully reviewed all suggested modifications, making final editorial decisions to ensure the scientific accuracy and integrity of the content remained unaltered. The intellectual contributions, including all experimental design, analysis, and conclusions, are solely those of the human authors.