# OpenReview forum: "KG-MoE: Multimodal Knowledge Graph Grounded Mixture of Experts for Fair Visual Question Answering"
_ICLR.cc/2026/Conference — Submitted to ICLR 2026_

### Official Review · Reviewer_B1YP · 2025-10-28

**Soundness:** 2
**Presentation:** 1
**Contribution:** 1
**Rating:** 0
**Confidence:** 3

**Summary:**

The paper proposes KG-MoE, a multimodal Mixture-of-Experts that conditions expert routing and explanation on retrieved knowledge-graph (KG) subgraphs and trains with fairness-aware objectives (worst-group risk + adversarial invariance). It claims theoretical guarantees that KG-conditioned routing improves worst-group risk under shift and that fairness regularization tightens a worst-group excess-risk bound.

**Strengths:**

The primary strength is in the integration of a KG into the MoE routing mechanism. Instead of just using a KG as an external information source post-prediction, this work uses it to guide the model's internal processing, forcing expert selection to be consistent with structured domain knowledge.

**Weaknesses:**

- A major discrepancy exists between the paper’s stated contributions and its empirical evaluation. The manuscript claims that the framework “outputs both predictions and knowledge-grounded explanations,” presents “qualitative analyses that show knowledge-aligned explanations mirroring clinical reasoning,” and evaluates explanation quality using BLEU, ROUGE, BERTScore, and a grounding rate. Yet neither the main paper nor the appendix reports the actual results for these metrics. This omission directly undermines the paper’s headline contribution, Multimodal Knowledge Graph Grounded Mixture of Experts for Fair Visual Question Answering. Notably, VQA/explanation quality is not evaluated at all. In its current form, the core claim about its contribution remains unverified and is materially misleading.
- The method assembles a large, hand-crafted ensemble of specialized experts across modalities (clinical photos, dermoscopy, histopathology) and problem facets (fairness, OOD etc.,), each trained with different architectures, datasets, and losses. This raises several issues (i) It’s hard to isolate the gains from KG-conditioned routing versus the gains from using a large ensemble; (ii) How feasible is this design in new domains that lack the resources to build many tailored experts?
- While Group-DRO and IRM are classic, recent subgroup-robust and distribution-shift methods are under-discussed and untested.

**Questions:**

- See weaknesses.

---

### Official Review · Reviewer_wAAs · 2025-10-30

**Soundness:** 2
**Presentation:** 2
**Contribution:** 2
**Rating:** 4
**Confidence:** 2

**Summary:**

The paper introduces KG-MoE, a knowledge-grounded and fairness-aware Mixture-of-Experts framework. It employs a probabilistic gating function parameterized over joint feature embeddings and retrieved subgraph representations. To ensure fairness, the method integrates worst-group risk minimization and adversarial invariance objectives, supported by theoretical analysis. Experiments on multimodal benchmarks demonstrate the model’s effectiveness.

**Strengths:**

1. A detailed ablation study presented in Table 2 demonstrates the effectiveness of the proposed key components.
2. Theoretical analysis is provided to demonstrate how knowledge grounding reduces excess risk under distributional shifts, while fairness regularization enhances worst-group generalization.
3. A comprehensive evaluation is conducted across multiple datasets.

**Weaknesses:**

1. The paper is somewhat difficult to follow due to the presence of numerous equations without adequate intuitive explanations. For instance, in Equation 12, the load balance and soft capacity penalty loss terms are introduced without clarifying their purpose or impact. Similarly, the coarse- and fine-grained losses in Equation 18 are not defined. The authors should at least explain the meaning and motivation behind each term to improve readability. Furthermore, the rationale for selecting τ = 0.7 and K = 13 immediately after Equation 15 is not provided.
2. In Equation 15, it is unclear how the modality-specific visual embeddings, meta embeddings, and subgraph embeddings are integrated into the gating function. Providing additional details on this process would greatly enhance clarity.
3. In Table 1, the proposed technique performs noticeably worse than the standard single-modality models, particularly on the Pathology dataset. Providing an explanation or justification for this performance gap would be helpful.
4. The impact of key components such as the load balancing and soft capacity losses on performance is not demonstrated. Including an ablation study analyzing their respective hyperparameters would be valuable. Furhermore, examining the effect of the coarse- and fine-grained losses from Equation 18 would further strengthen the analysis.

**Questions:**

Please refer to Weaknesses Section

---

### Official Review · Reviewer_ELPJ · 2025-10-30

**Soundness:** 3
**Presentation:** 2
**Contribution:** 2
**Rating:** 6
**Confidence:** 3

**Summary:**

This paper proposes KG-MoE, a novel Mixture-of-Experts framework that integrates structured knowledge graphs for routing and inference while incorporating fairness-aware objectives. The method is rigorously evaluated in the high-stakes domain of dermatology, leveraging a large-scale, multimodal dataset (200k+ images, 1.5M VQA pairs). The core idea is to use retrieved KG subgraphs to condition a gating network, thereby aligning expert specialization with knowledge-informed context. The authors provide a substantial theoretical analysis, deriving bounds on excess risk under distribution shift and worst-group generalization. Empirically, KG-MoE achieves state-of-the-art performance across clinical, dermoscopic, and histopathology tasks while significantly reducing the Fitzpatrick skin tone parity gap by over 50% compared to strong baselines. The supplementary material greatly strengthens the paper with full theoretical proofs, detailed dataset construction, and extensive ablations.

**Strengths:**

1. The method systematically integrates the knowledge graph into the MoE routing and reasoning process.
2. The paper provides a complete theoretical analysis, including routing stability, knowledge superiority theorem, and worst group risk upper bound, providing a solid mathematical foundation for the method.

**Weaknesses:**

1. The knowledge graph used is statically constructed and does not consider the dynamic update mechanism of medical knowledge, which may limit its long-term applicability.
2. The assessment dimension of fairness remains relatively simple. Although the method supports multi-attribute fairness, the experimental assessment still mainly focuses on skin color, and there is insufficient evaluation of other attributes (such as age, gender, region) and their cross-combinations.
3. Lack of forward-looking and clinical deployment validation. The experiment was based on retrospective data and lacked forward-looking validation in a real clinical setting, without collaboration with doctors.

**Questions:**

1. Is there a plan to introduce a dynamic knowledge update mechanism (such as based on the latest medical literature or clinical guidelines) to maintain the timeliness of the knowledge graph?
2. Have internal experiments been conducted on other attributes (such as age, gender) or cross-groups? What are the results?
3. How does the routing mechanism maintain stability when knowledge graph retrieval fails or returns noisy information? Is there any analysis of related failure modes?

---

### Official Review · Reviewer_B1bp · 2025-11-03

**Soundness:** 3
**Presentation:** 1
**Contribution:** 3
**Rating:** 4
**Confidence:** 3

**Summary:**

The paper proposes KG-MoE, a Mixture-of-Experts framework for dermatology that uses knowledge graphs to route between specialized experts and includes explicit fairness optimization to reduce disparities across skin tones. The method retrieves relevant medical knowledge from a dermatology knowledge graph (DermKG) to inform which experts should handle each case, and applies worst-group risk optimization with adversarial debiasing to improve fairness.
The paper evaluates on multiple dermatology datasets covering three modalities (clinical photos, dermoscopy, histopathology). Results show that KG-MoE achieves competitive or better accuracy than individual expert baselines across all modalities, and reduces the Fitzpatrick skin tone disparity gap by over 50% compared to baselines.

**Strengths:**

* High-stakes real-world application (Dermatology) is an important medical domain. The paper demonstrates quantitative improvements: MoE achieves competitive or superior performance across all three modalities (clinical, dermoscopy, pathology) compared to individual expert baselines.

* Comprehensive multi-dataset evaluation using multiple diverse public datasets, spanning different modalities, populations, including fairness-focused datasets and external validation sets for testing generalization.

**Weaknesses:**

* The paper frames the problem as VQA but provides insufficient details about the question-answer pairs: no information on how questions were gathered, question types, answer space size, or illustrative examples. Critically, the VQA dataset is synthetically generated by VLMs, which raises serious validity concerns for training a medical diagnostic system on generated rather than real annotated data.

* Figure 1 is poorly designed with confusing and unclear information flow: arrows don't clearly show the pipeline from inputs to outputs, the connection between expert outputs and the VLM is ambiguous, notation is inconsistent (what is the distinction between E1 in the inner rectangle vs. E1 in outer rectangle? same for Em), and training components (Pretrain Experts, Fine-tune, Adversarial Optimization) are mixed with inference flow without clear separation. It's unclear what exactly feeds into "Predictions" vs. "Rationales," how the "VLM + KG Constraints" box integrates with expert predictions. The figure makes the core architectural contribution difficult to understand.

* The distinct roles and outputs of experts vs. VLM are poorly explained. the text is ambiguous about whether the VLM contributes to diagnostic predictions or only post-hoc explanations. What are the outputs of experts and VLM specifically? Additionally, no examples of VLM-generated rationales are shown.

* Tables report top-2 and top-3 fusion, but it's unclear whether this refers to: (1) selecting and fusing top-k expert predictions (and if so, how predictions from different class spaces—e.g. 40 dermoscopic classes vs. 16 clinical classes are fused), or (2) weighted combination of expert representations per eq. 20 followed by a shared classifier. The paper must explicitly clarify the fusion approach.

* Tables lack explicit references to baselines (citations missing for baselines).

* Expert configuration lacks clear justification. paper provides no justification for: Why these specific expert types were chosen?
What makes each expert "specialized" beyond modality? (e.g., what does the "Fairness Expert" specifically learn?). How the number of experts was determined? overall, what justifies the expert configuration as studied in the paper? Although ablation study comparing full 9-expert configuration vs. 3 & 5 expert configurations, show better performance, the specific chosen expert configuration seems arbitrary. The presented table only shows that more experts is better, not that the specific expert configuration is optimal. The paper would particularly benefit from a "Leave-one-out" ablation showing the contribution of each expert type. A qualitative analysis of expert activations i.e. which experts activate for different input scenarios (e.g., dark vs. light skin, rare vs. common diseases, clear vs. ambiguous cases) could provide further insights not only about the value of each expert, but also about the routing decisions.

* In Figure 3 of supplementary material, Image quality is poor with unreadable text labels.

* In table 1, single-modality baselines achieve higher Macro-F1, AUROC, and CS-F1 than MoE on Pathology, yet MoE results are boldfaced. Bold formatting should distinguish best performance only, with second-best indicated separately if needed.

Overall the paper is difficult to understand due to unclear explanations of the core architecture and method. Key components like the fusion mechanism, the role of the VLM vs. experts, and the information flow are not properly explained. Additionally, the reliance on synthetically generated training data (created by other VLMs rather than real clinicians) raises serious concerns for a medical diagnostic system, and the clinical validation is inadequate with minimal details on how expert evaluation was conducted. These clarity and validation issues make it hard to properly assess whether the method is sound and suitable for medical use.

**Questions:**

Refer to weaknesses

---

### Meta-Review · Area_Chair_FYtD · 2026-01-07

**Summary:**

Reviewers raised concerns over motivation, fairness in experiments, and multiple aspects in presentation. The authors did not respond to them.

**Reviewer Concerns:**

`B1bp`: missing key rationales and benchmark details; multiple issues with presentation; lack of key justifications in the experiments

`ELPJ` static KG, fairness concerns in experiment setups, lack of clinical validation

`wAAs`: multiple issues with clarity in presentation, missing rationales for some design choices, insufficient ablations on key components.

`B1YP`: discrepancy between claimed contributions and experiment validations, fairness concerns in comparison and lack of rationales, robustness/distribution shift under-discussed.

The authors did not respond to the questions.

**Reviewer Scores:**

The authors did not respond to the questions. Reviewers are therefore unlikely to change their scores.

---

### Decision · Program_Chairs · 2026-01-26

Reject